# Can Psychological Empowerment Prevent Emotional Disorders in Presence of Fear of COVID-19 in Health Workers? A Cross-Sectional Validation Study

**DOI:** 10.3390/jcm10081614

**Published:** 2021-04-10

**Authors:** Marta Llorente-Alonso, Cristina García-Ael, Gabriela Topa, María Luisa Sanz-Muñoz, Irene Muñoz-Alcalde, Beatriz Cortés-Abejer

**Affiliations:** 1Health Psychology Program, International School of Doctorate, National Distance Education University (UNED), C/Bravo Murillo, 38, 3ª, 28015 Madrid, Spain; 2Gerencia de Asistencia Sanitaria del Área de Salud de Soria, Complejo Hospitalario de Soria, Gerencia Regional de Salud de Castilla y León (Sacyl), Pº Santa Bárbara s/n, 42005 Soria, Spain; mlsanzm@saludcastillayleon.es (M.L.S.-M.); ima87312@gmail.com (I.M.-A.); becorabe@gmail.com (B.C.-A.); 3Department of Social and Organizational Psychology, Faculty of Psychology, National Distance Education University (UNED), C/ Juan del Rosal, 10, 28040 Madrid, Spain; cgarciaael@psi.uned.es (C.G.-A.); gtopa@psi.uned.es (G.T.)

**Keywords:** psychological empowerment, fear of COVID-19, collaborative crafting, job crafting, emotional disorders, psychological detachment

## Abstract

The global emergency produced by COVID-19 has been a turning point for health organizations. Healthcare professionals have been exposed to high levels of stress and workload. Close contact with infected patients and the infectious capacity of COVID-19 mean that this group is especially vulnerable to contagion. In various countries, the Fear of COVID-19 Scale has been shown to be a fast and reliable tool. Early detection of fear complements clinical efforts to prevent emotional disorders. Thus, concepts focused on positive occupational health, such as Job Crafting or psychological empowerment (PE), have been examined as a tool to prevent mental health problems at work. In this work, we intended to adapt and validate the 7-item Fear of COVID-19 Scale in health workers (*N* = 194). The interpretation of the measurement model indicates adequate values of internal consistency reliability, and convergent and discriminant validity. The overall goodness of fit of the model was also adequate. The structural model indicates that the implementation of job crafting measures in health services leads to workers’ greater PE. High levels of anxiety and depression prevent health professionals from psychologically detaching from work. In turn, PE can reduce the emotional disorders caused by the fear of COVID-19.

## 1. Introduction

The health emergency caused by the SARS-COV-2 virus has led to serious physical and psychological problems worldwide [1,2]. In Spain, the “Center for Sociological Research” (October 2020) [3] reports that 79.3% of the Spanish population considers that the pandemic has affected the emotional health of the entire population. This survey was conducted on 2861 people and it values the effects and consequences of the Coronavirus in the Spanish population. Of the participants, 50.6% expressed anxiety during the health crisis, 29.3% felt depressed, and 57.5% were afraid of getting sick. In addition, subsequent surveys have shown that the pandemic has changed the way a large part of Spanish society thinks. In fact, 12.3% of the respondents consider that they live in fear, unease, or apprehension of the pandemic (CIS, December 2020) [4].

The high contagion rate and increased mortality of the SARS-COV-2 virus compared with other respiratory pathologies [5] have caused feelings of fear and uncertainty about the future in part of the population [4]. Fear consists of anguish over a real or imaginary risk or harm [6]. Extreme fear has even led to cases of suicide in people not diagnosed with COVID-19 [7,8]. Gunnell et al. suggest that the COVID-19 pandemic can trigger profound effects on mental health, and that suicide rates may increase, given the increase in the number of suicides in previous epidemics (in the USA during the 1918–1919 flu and among older people, in Hong Kong during the 2003 Severe Acute Respiratory Syndrome (SARS) epidemic) [9]. Therefore, fear assessment can be an important mechanism for preventing emotional or mental health disorders.

The general objective of this study is to adapt and validate the Fear of COVID-19 Scale of Ahorsu et al. [10], made up of 7 items, in health workers. This research also comprises several specific objectives. First, we aim to examine the role that fear of COVID-19 plays in emotional disorders and, in turn, in psychological detachment. Second, we will assess whether positive occupational health, through collaborative crafting and psychological empowerment (PE), can act as a relevant factor in preventing emotional disorders and lead to better recovery experiences after the workday. Finally, we intend to determine whether the fear of COVID-19 has any negative effect on PE and, in turn, whether it affects how healthcare professionals’ distance themselves from work.

### 1.1. Current Situation of the Fear of COVID-19

Ahorsu et al. developed a brief instrument to detect fear of COVID-19 in the general population [10]. As these authors explained, fear is directly associated with the transmission rate and morbidity. This scale has been adapted to other cultures in general population [11,12,13,14,15,16,17,18,19]. However, no psychometric adaptations and assessments of the Fear of COVID-19 Scale in health workers were found in the literature. The assessment of fear of COVID-19 levels in different sociodemographic groups is relevant for the implementation of specific prevention programs [8]. We also consider it especially important to know whether there is fear in professional sectors that are in direct contact with the virus and that have a high risk of exposure contagion. Early detection of fear of COVID-19 can act as an alarm signal to health workers to prevent the development of emotional disorders and, therefore, be able to recover from stressful work situations.

On the other hand, a recent meta-analysis showed that health professionals working to combat COVID-19 are more affected by psychiatric disorders, sleep disorders, stress, and indirect trauma than other occupational groups [20]. Pappa et al. Additionally suggested that a significant proportion of health workers have experienced mood and sleep disorders [21]. De Brier et al. stated that the level of exposure to the disease and fear for health were significantly associated with worse mental health outcomes [2]. Additionally, finding oneself in stressful, scary situations can lead to emotional disorders that, in turn, can prevent proper psychological detachment from work. This detachment is part of a process called recovery, through which people stop facing a demanding situation to regain energy to continue and renew the resources invested in that situation [22]. Stressful experiences are considered the opposite process of recovery [22]. Sonnentag et al. reported that employees who felt exhausted found it more difficult to disconnect psychologically from work [23]. They also stressed the importance of time pressure in the increase of the association between exhaustion and lower psychological detachment. In this sense, the duration of stressful experiences and feelings of fear during the COVID-19 pandemic can contribute to health workers’ being unable to disconnect from work, thus preventing adequate recovery experiences.

### 1.2. Job Crafting and Psychological Empowerment

Job crafting is defined as employee’s proactive behavior aiming to modify the relational, cognitive, or task limits to shape or redesign a job [24]. Most research has studied job crafting on an individual level. However, Leana et al. consider job crafting not only to consist of an individual employee’s activity, but of the fact that workers participate in similar work processes, relate to each other, and experience common events [25]. This refers to collaborative crafting, whereby employees can team up and decide how to modify tasks to achieve their goals. Moreover, in certain professions, such as in health care or education, it is difficult to adapt individual work due to the high degree of interdependence between groups [26]. Research linking job crafting to PE is still scarce. However, several authors have shown that job crafting is strongly related to PE [27,28]. Specifically, Harbridge conducted a study on registered nurses and highlighted job crafting as an important predictor of PE [28]. Demerouti proposed that job crafting may lead to greater motivation, performance, or engagement [29].

On the other hand, PE consists of a subjective, cognitive, and attitudinal process through which the individual feels effective, competent, and authorized to perform tasks. Spreitzer considers that PE reflects an active orientation and self-perception of the ability to shape one’s working role [30]. While PE is not synonymous with intrinsic motivation, it can be considered a predictor of it [31], and therefore a motivational factor. The four components of PE are a proximal cause of intrinsic task motivation and satisfaction [31]. Schermuly and Meyer showed that PE leads to less emotional fatigue and depression [32]. It also strongly influences the degree of work stress experienced by workers [33]. Petersen et al. found evidence that self-efficacy is amendable to change and exerts an effect on protective behavior. The effects of fear were small among those who felt efficacious [34].

On the other hand, Ghosh et al. found that psychological detachment acts as a moderator between intrinsic motivation and engagement [35]. Employees who feel motivated and psychologically detached from work in their free time are also more creative [35].

Thirdly, to test the last specific objective, we aim to determine whether fear of COVID-19 has any negative effect on psychological detachment, through the mediation of PE. There is no literature linking the fear of COVID-19 to organizational variables such as PE. However, other types of fear, such as fear of success, have been linked to self-efficacy and intrinsic motivation. Specifically, both of them can be used to mitigate the potentially adverse effects of this type of fear [36]. In this sense, in a situation of fear of COVID-19, workers are expected to have reduced PE and be incapable of activating the cognitive processes that enable them to perform tasks effectively and competently. In this way, they will not be able to distance themselves from their work in their free time.

**Hypothesis** **1.***Emotional disorders will play a mediating role in the relationship between the fear of COVID-19 and psychological detachment*. 

**Hypothesis** **2.***PE will mediate the relationship between collaborative crafting and emotional disorders*.

**Hypothesis** **3.***Emotional disorders will mediate between PE and psychological detachment*. 

**Hypothesis** **4.***PE will mediate the relationship between fear of COVID-19 and the psychological detachment of health professionals*.

## 2. Materials and Methods

### 2.1. Participants and Procedure

The sample was made up of a total of 194 workers from the health centers of the province of Soria belonging to the Health Service of Castilla y León (Sacyl) in Spain. Permission was requested from the organization’s Ethics Committee, and the questionnaires were forwarded to a total of 1056 workers, obtaining a response rate of 18.37%. Data collection took place in July 2020 via email. Through this means, participants accessed a link in Google Forms by which, after providing their informed consent, access was given to fill out the questionnaire. The final sample consisted of 162 women and 32 men, with an average age of 45.94 (SD = 12.39). Of the sample, 28.4% had been diagnosed with COVID-19 or had been detected to have antibodies after a test. Concerning their employment status, 50.5% of the participants were nurses or specialist nurses, 26.3% were specialized graduates, and 12.4% were assistant nursing technicians. Hence, most of the participants were healthcare professionals. Additionally, 45.9% of the sample considered that their tasks or work activities during the COVID-19 pandemic had changed compared with those they had performed previously. (see Table 1).

### 2.2. Instrument

To assess fear of COVID-19, the Fear of COVID-19 Scale was used [10]. As a preliminary step, the questionnaire was translated into Spanish by a blind back-translation process [37]. Two methodology experts compared the original scale and the final Spanish version [38]. The questionnaire is a one-dimensional scale consisting of seven items measured on a Likert-like response scale ranging from 1 (Totally Disagree) to 5 (Totally Agree). The internal consistency index of the scale (α = 0.90) was higher than that obtained for the original scale (α = 0.82) [10]. 

As mentioned above, PE refers to several cognitive processes that modify the subjective self-perception, by which the worker feels intrinsically motivated and effective to perform tasks [30]. In this study, PE was evaluated using the Psychological Empowerment Scale [30], adapted to Spanish [39] consisting of four subscales: (a) Meaning (three items, e.g., “My work activities have been personally valuable”); (b) Competence (three items, e.g., “I trust my ability to get the job done”); (c) Self-determination (three items, e.g., “I have had the autonomy to determine how to do my job”); and (d) Impact (three items, e.g., “I’ve had enough influence on what was going on in my work”). The Likert-type response scale ranged from 1 (Totally Disagree) to 5 (Totally Agree). High scores indicate greater PE. The reliability coefficient was high (α = 0.84).

To assess participants’ anxiety and depression, we used the Hospital Scale of Anxiety and Depression (HADS) [40], consisting of two subscales of seven items each. Items in the Anxiety subscale aim to detect generalized anxiety (“I feel tense or nervous”), and the subscale of Depression primarily assesses the state of anhedonia (“I feel like I’m slowing down every day”). The response range ranged from 1 (Totally Disagree) to 5 (Totally Agree). The internal consistency in this study was high for both the total scale (α = 0.92), the Anxiety subscale (α = 0.91), and the Depression subscale (α = 0.87).

To evaluate collaborative crafting, the two-dimensional Spanish-validated Job Crafting Questionnaire [25,26] was used. In the study, we used the 6-item Collaborative Crafting Subscale, of which only five items were used (e.g., “You work together with your peers to introduce new approaches to improving your work”). Specifically, the item that refers to celebrations or events at work was removed, as it was deemed inappropriate in the pandemic situation. The response scale ranged from 1 (Totally Disagree) to 5 (Totally Agree). The reliability coefficient was high (α = 0.91).

To measure psychological detachment, we used the Recovery Experience Questionnaire [41] validated in Spanish [22], which presents four subdimensions: Psychological Detachment, Relaxation, Challenge-Seeking, and Control. For the study, three items of the Psychological Detachment subscale were used (e.g., “After work, I can disconnect) measured on a Likert-type scale ranging from 1 (Totally disagree) to 5 (Totally agree). The reliability coefficient was high (α = 0.95).

Finally, the following demographic data were collected: age, gender, professional category, organizational rank, job tenure in the current contract, diagnosis of Covid, place of work, type of contract, and modification of tasks or activities during the pandemic.

### 2.3. Data Analysis

To evaluate the descriptive statistics and bivariate correlations of the study variables, we used the IBM SPSS Statistics 26 program [42]. The data was then analyzed using a structural equations model (SEM) based on the variance, with the method of partial least squares (PLS) [43]. This procedure allows simultaneously assessing the reliability and validity of the measures of the theoretical construct (measurement model) and estimating the relationships between constructs (structural model) [44]. A new approach called consistent PLS was used because, if the common factor model is retained, consistent PLS or covariance-based SEM should be the first choice of researchers over traditional PLS [45]. Additionally, when comparing PLS-SEM with CB-SEM, PLS can handle small sample sizes and discard the assumption of normality, so it is recommended for social science research [46]. Thus, in the present investigation the method of choice is PLS-SEM, which is considered a more robust method when the sample size is reduced. The data were analyzed with the statistical software SmartPLS (v.3.3.2) [47].

## 3. Results

The mean, standard deviations, and correlations of the study variables are presented in Table 2.

Before analyzing the data for the adaptation of the Fear to COVID-19 scale, we examined skewness and kurtosis to verify that the data did not stray excessively from a normal distribution. PLS-SEM is a non-parametric statistical method [48]. Although the data are not required to have a normal distribution, if extremely non-normal data were present, the standard errors obtained through bootstrapping could be inflated, and the probability of finding significant relationships between variables would decrease [48]. These authors recommend examining two distribution measures, skewness and kurtosis. In most indicators, values between −0.99 and +0.98 were obtained, so we decided not to eliminate any of them, as there was no problem of non-normality.

The interpretation of the PLS model comprises three phases: (a) evaluation of the global model, (b) measurement model (external model), and (c) structural model (internal model). We also performed an analysis of the invariance of the measurement model to determine whether similar measures were obtained in different groups.

### 3.1. Global Model

To evaluate the global model, the Standardized Root Mean Square Residual (SRMR) parameter was used, which measures the difference between the observed correlation matrix and the correlation matrix implied by the model. Hu and Bentler proposed values of SRMR < 0.08 to achieve a good fit of the data [49]. Ringle proposed a more flexible option (SRMR < 0.10) [50]. In this study, we obtained an SRMR of 0.077 both in the saturated and estimated models.

### 3.2. Measurement Model

The evaluation of the reflective measurement models includes composite reliability (to assess internal consistency), the reliability of the individual indicator, and the mean-variance extracted (AVE) to assess convergent validity [48]. In addition, such measurement models also assess discriminant validity.

First, the internal consistency reliability of the Fear of COVID-19 Scale was tested. Cronbach’s alpha coefficient obtained a high value of 0.90. Composite reliability and reliability rho_A obtained values of 0.89 and 0.90, respectively. These composite reliability values are considered satisfactory. Values greater than 0.95 are not adequate because they could suggest that all indicators are measuring the same phenomenon [48]. The rest of the variables also obtained high levels of internal consistency.

Second, convergent validity was assessed, examining the loadings or simple correlations of indicators with their construct. The external loadings of the indicator should be greater than 0.707 [51]. In the case of our Fear of COVID-19 Scale, items 5, 6, and 7 exceeded the value of 0.707, whereas the rest obtained values between 0.58 and 0.67. Indicators with loadings between 0.40 and 0.70 should be removed if there is an increase in composite reliability [52]. After performing the analyses without these items, the composite reliability only decreased from 0.89 to 0.84, so we decided to maintain them. For all other variables, several analyses were performed, eliminating indicators with values between 0.40 and 0.70. In the case of emotional disorders, there was an increase in composite reliability (0.92) and AVE (0.50), so item 4 was removed from the Anxiety scale and item 4 from the Depression scale. On the PE scale, items 1 and 2 (Meaning subscale), 5 (Competency subscale), 7 (Self-Determination subscale), and 10 (Impact subscale) were also removed. Thus, a high composite reliability value (0.84) was obtained.

Convergent validity was evaluated through AVE. The Fear of COVID-19 Scale obtained a value of 0.55, above the recommended value of 0.50. This indicates that the construct explains more than half of the variance of its indicators. The rest of the variables, except for PE, achieved values greater than 0.50.

Finally, discriminant validity was assessed through the cross-loads, following the criterion of Fornell and Larcker and the Heterotrait-Monotrait ratio (HTMT). The load of the indicators on the Fear of COVID-19 Scale was higher than their cross-loads with other constructs, indicating discriminant validity. Concerning the Fornell criterion, the square root of the AVE of the Fear of COVID-19 (0.75) was not higher than the correlation between fear and emotional disorders (0.87). However, given the absence of discrepancies between the two criteria, we valued the HTMT. The Fornell and Larcker criterion is not appropriate when the loadings of the indicators of the constructs differ only slightly [48]. HTMT should be lower than 0.85. All the variables had lower values, so discriminant validity was achieved (see Table 3) [53].

### 3.3. Structural Model

Having verified that the measures of the constructs are reliable and valid, we valued the structural model. First, we evaluated the collinearity of the structural model, using the variance inflation factor (VIF) whose value must be 5 or less [52]. The results showed that all VIF values were below 5, indicating the absence of collinearity between predictors. Specifically, the VIF value between EP and Collaborative Crafting, and between EP and fear of COVID-19 was 1.01. Between emotional disorders and PE, and emotional disorders and fear, the VIF was 1.05, whereas between PE and psychological detachment, it was 1.12. The highest values of VIF were 4.53 and 4.30, between detachment and emotional disorders, and detachment and fear, respectively.

The algebraic sign, magnitude, and statistical significance of the path coefficients were also evaluated. The signs of the path coefficients matched the hypotheses raised. The highest values of the standardized beta coefficients (β) were between fear of COVID-19 and emotional disorders (β = 0.85, *p* < 0.001) and between emotional disorders and psychological detachment (β = −0.82, *p* < 0.001).

Bootstrapping was used for consistent PLS (10,000 subsamples) to assess the meaning of the path coefficients. The relationships between Collaborative Crafting and PE (*t* = 7.90, *p* < 0.001), between PE and emotional disorders (*t* = 2.28, *p* = 0.023), between fear of COVID-19 and emotional disorders (*t* = 24.01, *p* < 0.001), and between emotional disorders and psychological detachment (*t* = 3.95, *p* < 0.001) were significant. In contrast, fear of COVID-19 was not directly related to psychological detachment (*t* = 1.24, *p* = 0.21), or to PE (*t* = 1.90, *p* = 0.05). 

On the other hand, we calculated the indirect effects between the variables. The indirect effect of fear of COVID-19 on psychological detachment (β = −0.69, *p* < 0.001), through the mediation of emotional disorders, was significant. The indirect effect of collaborative crafting on emotional disorders was also significant, through the mediation of PE (β = −0.06, *p* = 0.033). Additionally, the mediation of emotional disorders in the relationship between PE and psychological detachment was significant (β = 0.10, *p* < 0.001). These results support Hypotheses 1, 2, and 3. However, Hypothesis 4 could not be confirmed. In Figure 1, we specify the structural model. In Table 4, we specify the total effects of the model.

With regard to the coefficient of determination, the model explained 40.9% of the variance of psychological detachment, 77.9% of the variance of emotional disorders, and 30.6% of PE.

### 3.4. MICOM Model: Analysis of the Invariance of the Measurement Model

Measurement invariance, or measure equivalence, means that group differences in model estimates are not due to the different content or meaning of the latent variables between groups [48], but as to whether, under different conditions of observation and study of the phenomena, the measurement operations produce measurements of the same attribute [54]. Henseler et al. developed a procedure for calculating the measurement invariance of composite models (MICOM) [55]. This method is developed in three hierarchically interrelated stages. As Henseler et al. explain, variance-based SEM techniques model latent variables as composite variables, so this procedure is considered appropriate for assessing common factor models, such as the one presented in this study [55].

#### 3.4.1. First Stage: Configuration Invariance

This determines whether a composite has been specified equally in all groups and whether it emerges as a one-dimensional entity in the same nomological network for all groups [55]. In this study, the initial qualitative evaluation ensures that the same indicators are used in each measurement model, the data are processed in the same way, and the algorithm is also configured identically.

#### 3.4.2. Second Stage: Composite Invariance

It analyzes whether a composite is formed in the same way in all groups [55]. To evaluate composite invariance, we performed a permutation algorithm with PLS (5000 permutations), in which the selected groups were, on the one hand, the participants whose work tasks or activities had been changed during the COVID-19 pandemic, and on the other hand, those whose activity was similar to before the health emergency. To test composite invariance, the original correlation must be greater than or equal to the 5% quantile. In Table 5, MICOM results show that the composite scores did not differ between the two groups.

#### 3.4.3. Third Stage: Evaluation of Equal Means and Variances of the Composite Variables 

In the third step, we determined whether the original differences in means and variances were between 2.5% and 97.5%, and complete invariance was established. If only one of these variances fell between 2.5% and 97.5%, then partial invariance of means and variances would be considered. In Table 6, the results of the third MICOM stage suggest equality of means and variances.

Therefore, after observing that all three stages were met, we performed multigroup analysis based on group data [48]. We ran the multigroup analysis in PLS and analyzed the non-parametric PLS-MGA approach, which compares each bootstrap estimate of a given parameter between each of the groups [48]. Again, we performed the calculation with 5000 subsamples. All values were nonsignificant, indicating that no bootstrap estimate of a parameter differed between groups. As can be seen in Table 7, our results suggest that there are no significant differences between the two groups (changes in work activities vs. no changes in work activities).

## 4. Discussion

The main objective of this work was to adapt and validate the Fear of COVID-19 Scale [10], composed of seven items, in health workers. The results show that it is a valid and reliable scale and that its structure consists of a single factor, as other authors have suggested [10,12,13,15,16,18]. First, the evaluation of the overall model based on the SRMR criterion showed adequate goodness of fit, thus indicating that the model is probably appropriate. Second, by rating the external or measurement model, we conclude that both the composite reliability of the scale and the internal consistency reliability are adequate. Appropriate values of convergent and discriminant validity were also obtained. Other authors have found similar results in terms of composite reliability [10], internal consistency [13], convergent validity [10], and discriminant validity [56]. Finally, to ensure that the differences between groups are not due to the content or meaning of the latent variables, we tested the measurement invariance of composite models (MICOM). The results showed that there were no differences between the groups of participants (with and without changes in their tasks or activities), indicating the existence of measurement equivalence between the two groups.

The evaluation of the structural model determines whether the postulated assumptions are met. Concerning the first hypothesis, this research was intended to assess the role of fear of COVID-19 in the development of anxious-depressive disorders, and in health workers’ ability to distance themselves psychologically from their work. The results have shown that fear of COVID-19 is a strong predictor of emotional disorders, such that health professionals who score higher in fear of COVID-19 are more likely to develop anxiety and/or depression. Ahorsu et al. also found significant relationships between fear and anxiety or depression, suggesting that people with severe fear may have these emotional disorders [10]. Additionally, in this study, we found that fear determines health workers’ inability to distance themselves from work, but only if they suffer some degree of comorbid anxiety and depression. This result is in line with the results of Sonnentag et al., because health workers who feel more exhausted find it more difficult to detach psychologically from work [23]. 

Concerning the second hypothesis, which sought to test the ability of positive occupational health to prevent mental health problems at work, we found that collaborative crafting behaviors lead to health-care workers’ greater PE. In the first months of the pandemic, uncertainty due to the lack of knowledge of the coronavirus disease and the hospital collapse determined the need to work collaboratively and to modify the usual tasks of healthcare professionals (e.g., nurses or technicians who changed their jobs to work on Covid floors or doctors with non-COVID-19 specialties working together with internists or intensive care doctors). These collaborative crafting behaviors have proven to be important predictors of PE. Other authors have shown that PE leads to less emotional fatigue and depression [32]. In this study, we highlight the importance of PE as a mediator in the relationship between collaborative crafting and emotional disorders. These results provide evidence to the literature about the importance of job crafting interventions to enable employees to proactively create a motivating work environment and improve their well-being [57].

Next, we discuss the confirmation of the third hypothesis. We expected that the most empowered workers could psychologically distance more from work than those who are not empowered. In this relationship, we observed the importance of having no emotional disorders for health workers to recover after their working day, as anxiety and depression problems acted as a total mediator. Hochwälder and Brucefors also found that greater PE at work generally corresponds with fewer health problems [58].

Concerning the fourth hypothesis, which tested the role of fear in detachment through the mediation of PE, we found no significant relationships. Faced with a situation of fear of COVID-19, workers’ PE is not reduced, but other organizational variables, such as job crafting, activate the cognitive processes that enable them to perform their tasks.

Finally, we consider that one of the strengths of this research has been to relate positive organizational psychology to the prevention of basic emotions such as fear to decrease emotional disorders. Thus, we link lower fear scores with the preventive role of PE (after the development of collaborative crafting interventions) to develop fewer mental health problems and recover psychologically from work.

### 4.1. Limitations

Firstly, and as the main limitation, we highlight the impossibility of establishing relationships of direct causality. The findings should be interpreted with caution due to the cross-sectional nature of the data and the lack of longitudinal research on COVID-19 fear and emotional disorders. Secondly, the data are self-reported, so they should be treated with caution. Additionally, we consider a possible threat to external validity. When health workers responded to the questionnaire, they could react to the pandemic situation and respond according to the social norm, as a function of what is expected of the group of health professionals. For example, if a healthcare provider verbalizes that he or she is afraid of COVID-19, this may be criticized from the point of view of normative influence. Thirdly, as a threat to internal validity, we highlight the motivation to answer the questionnaire, or self-selection bias. The participants in this study may have had different expectations than those who chose not to participate. In future research focused on health workers, it would be advisable to assess the participants’ degree of social desirability.

On the other hand, this study was carried out in a single health institution and in the province with the highest incidence of seroprevalence in Spain [59]. This could pose a problem for the generalization of the results to the rest of health professionals in this country. In addition, another important limitation is the small size of the sample that can cause low representativeness. 

In addition, this study was carried out in July 2020, after three months of quarantine in Spain, and Soria was the most affected province, with a 14% seroprevalence of SARS-COV-2 [59]. However, the study was conducted at a time when the hospital situation was adequate. There was no hospital overload or collapse. In the questionnaire, professionals were instructed to evaluate the previous three months. However, these previous circumstances could have altered the answers.

Finally, it should be noted that these results were obtained in an exceptional situation of a global pandemic. Health workers were exposed to contagion due to lack of personal protective equipment, uncertainty, fear of infecting family members, helplessness from lack of knowledge about the disease, etc. Collaborative crafting was analyzed in a situation where health workers performed as a team and collaboratively more than ever. For this reason, we must be cautious and consider the exceptionality of the situation as a limitation of this study and analyze these organizational variables (job crafting and PE) in times of non-pandemic normality.

### 4.2. Future Lines of Research and Practical Implications

This study has important practical implications. We emphasize the importance of early detection of fear of COVID-19 to prevent emotional disorders, as well as to improve recovery experiences after the workday. Additionally, the Fear of COVID-19 Scale was adapted [10] for health workers, so it can be appropriately used to analyze this emotion in them. We also know the importance of promoting positive occupational health to prevent these anxious-depressive disorders. Through practices that enhance collaborative crafting, health workers can be empowered and thereby, reduce mental health problems. 

We found no research papers in the literature that relate positive organizational psychology and the psychology of emotions to mental health problems and recovery experiences. Coelho et al. recommended that future research on fear of COVID-19 and anxiety should focus on pointing to protective and risk factors of psychological well-being [60]. Other authors consider that leaders need to have the appropriate communication that provides up-to-date information and encourages individual empowerment to support their staff [61]. As future lines of research, more studies are proposed that investigate these models and analyze them over time through longitudinal research.

## Figures and Tables

**Figure 1 jcm-10-01614-f001:**
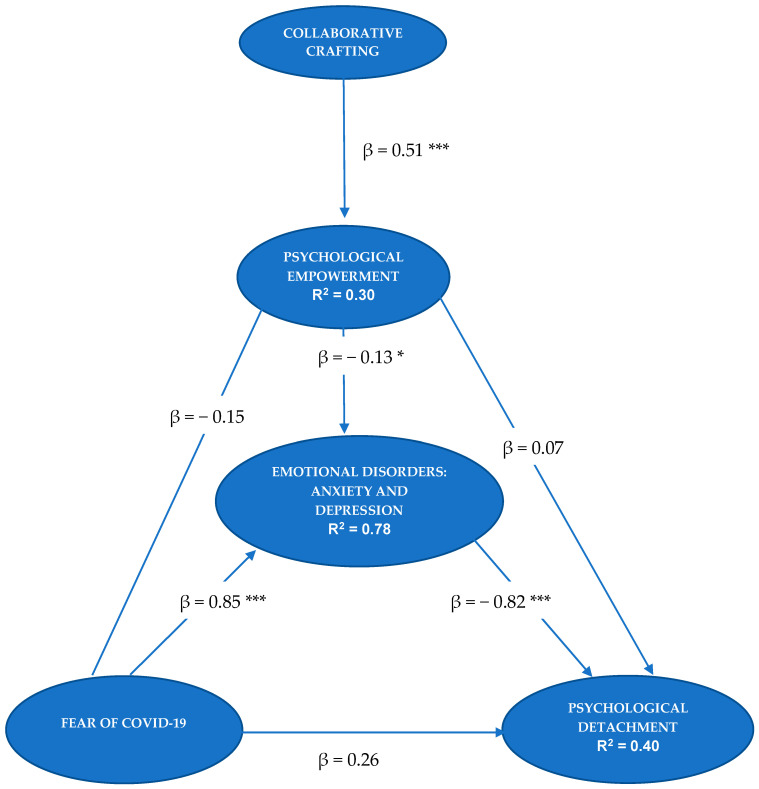
Structural model. * *p* < 0.05. *** *p* < 0.001.

**Table 1 jcm-10-01614-t001:** Demographics.

	*N*	%	*M*	*SD*
Age	194		45.94	12.39
Gender				
Female	162	83.5		
Male	32	16.5		
Professional category				
TCAE (nursing assistant)	24	12.4		
Nurse/Specialist Nurse	98	50.5		
Specialist graduate	51	26.3		
Administration	6	3.1		
Physiotherapy	7	3.6		
Social work	3	1.5		
Higher technician	5	2.6		
Organizational rank				
Intermediate or higher posts	46	23.7		
Workers without people in their care	148	76.3		
Job tenure in the current contract			12.20	12.97
Changing tasks or activities during the pandemic:				
Changes	89	45.9		
No changes	105	54.1		
COVID-19 diagnosis				
Yes	55	28.4		
No	139	71.6		
Type of contract				
Permanent	101	52.1		
Temporary	93	47.9		
Workplace during the pandemic				
COVID Floor or Team	83	42.8		
Non-COVID Service	46	23.7		
Health Center	57	29.4		
Telework/union release/administration	8	4.1		

Note: *N,* Sample Size; M, Mean; SD, Standard Deviation; TCAE, Technician in Auxiliary Nursing Care.

**Table 2 jcm-10-01614-t002:** Bivariate correlations, means, and standard deviations.

Variables	*M*	*SD*	1	2	3	4
Collaborative Crafting	3.63	0.97	-			
Psychological Empowerment	3.90	0.70	0.48 **	-		
Fear of COVID-19	2.38	0.90	−0.12	−0.16 *	-	
Emotional disorders	2.36	0.83	−0.17 *	−0.26 **	0.77 **	-
Psychological Detachment	2.92	1.20	0.16 *	0.20 **	−0.43 **	−0.59 **

Note: *N* = 302. * *p* < 0.05, ** *p* < 0.01. - indicates a blank space.

**Table 3 jcm-10-01614-t003:** Measurement model: loads, construct reliability, and convergent validity.

Latent Variable	Item	λ	CR	α	Rho_A	AVE
Collaborative Crafting	CC1	0.71	0.91	0.91	0.91	0.66
	CC2	0.84				
	CC3	0.68				
	CC4	0.86				
	CC5	0.94				
Psychological empowerment	PE3	0.51	0.84	0.84	0.85	0.44
	PE4	0.56				
	PE6	0.56				
	PE8	0.71				
	PE9	0.70				
	PE11	0.73				
	PE12	0.77				
Fear of COVID-19	F1	0.67	0.89	0.90	0.90	0.55
	F2	0.58				
	F3	0.66				
	F4	0.67				
	F5	0.89				
	F6	0.88				
	F7	0.81				
Psychological detachment	PD1	0.98	0.95	0.95	0.95	0.86
	PD2	0.91				
	PD3	0.88				
Emotional disorders	ED1	0.81	0.92	0.92	0.92	0.50
	ED2	0.84				
	ED3	0.74				
	ED5	0.77				
	ED6	0.71				
	ED7	0.82				
	ED8	0.62				
	ED9	0.63				
	ED10	0.67				
	ED12	0.63				
	ED13	0.57				
	ED14	0.60				

Note: λ = Loadings. CR = Composite reliability. Rho_A = Dijkstra-Henseler’s rho (ρA). AVE = Average variance extracted. Α = Cronbach’s alpha. Items removed: Psychological empowerment 1, 2, 5, 7, and 10; Emotional disorders 4 and 11.

**Table 4 jcm-10-01614-t004:** Total effects.

	Beta Coefficients	*t* Statistics	*p* Value
Crafting -> Detachment	0.091	2.228	0.026
Crafting -> Emocional Disorders	−0.064	2.132	0.033
Crafting -> Empowerment	0.513	7.863	0.001
Emocional Disorders -> Detachment	−0.822	3.975	0.001
Empowerment -> Detachment	0.177	2.404	0.016
Empowerment -> Emocional Disorders	−0.125	2.297	0.023
Fear -> Detachment	−0.468	8.114	0.001
Fear -> Emocional Disorders	0.866	27.97	0.001
Fear -> Empowerment	−0.152	1.912	0.056

**Table 5 jcm-10-01614-t005:** MICOM. Stage 2 results.

	Original Correlation	Correlation of Permutation Means	5.0%	*p*-Values of the Permutation
Collaborative Crafting	0.999	0.998	0.995	0.50
Psychological Detachment	1.000	1.000	1.000	0.95
Emotional Disorders	0.999	0.999	0.997	0.25
Psychological Empowerment	0.983	0.986	0.958	0.27
Fear of COVID-19	0.999	0.999	0.997	0.31

**Table 6 jcm-10-01614-t006:** MICOM. Stage 3 results. Original differences in mean and variance.

	Mean-Original Differences (Mean-Difference of Permutation Means)	2.5%	97.5%	*p*-Values of the Permutation	Variance-Original Difference (Variance-Difference of Permutation Means)	2.5%	97.5%	*p*-Values of the Permutation
Psychological Empowerment	0.18 (0.002)	−0.28	0.28	0.19	−0.18 (−0.004)	−0.42	0.43	0.40
Collaborative Crafting	0.18 (0.002)	−0.27	0.27	0.21	−0.10 (−0.004)	−0.36	0.36	0.61
Emotional Disorders	0.008 (−0.002)	−0.28	0.27	0.95	−0.07 (−0.004)	−0.34	0.32	0.67
Fear of COVID-19	−0.07 (−0.001)	−0.29	0.28	0.58	−0.07 (−0.001)	−0.42	0.39	0.72
Psychological Detachment	−0.18 (0.002)	−0.29	0.28	0.21	−0.02 (−0.002)	−0.27	0.27	0.88

**Table 7 jcm-10-01614-t007:** PLS-MGA (PLS-Multigroup Analysis results).

	Path Coefficients	Original 1-Tail *p*-Value	New *p*-Value
Crafting and Empowerment	−0.06	0.71	0.57
Emotional Disorders and Detachment	−0.16	0.78	0.42
Empowerment and Detachment	−0.18	0.94	0.12
Empowerment and Emotional Disorders	0.08	0.18	0.37
Fear and Detachment	0.33	0.05	0.11
Fear and Emotional Disorders	0.06	0.15	0.30
Fear and Empowerment	−0.04	0.61	0.77

Note: The contrasting groups were: Participants with changes in their tasks or work activities during the COVID-19 pandemic vs. participants with no changes in their tasks.

## Data Availability

The datasets generated and analyzed during the current study are available from the corresponding author on reasonable request.

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
