# Peer review of "Can Psychological Empowerment Prevent Emotional Disorders in Presence of Fear of COVID-19 in Health Workers? A Cross-Sectional Validation Study"

_jcm, 2021, doi:10.3390/jcm10081614_

Round 1

Reviewer 1 Report

The Article: Can Psychological Empowerment prevent Emotional Disorders 2 in presence of Fear of Covid-19 in Health Workers? A Cross-sectional Validation Study. Was submitted to: Journal of Clinical Medicine. In this manuscript, the authors examine 194 healthcare professionals with the 7-item Fear of Covid-19 Scale (Ahorsu et al., 2020). This work was not only used to adapt and validate the scale in this sample, but also to explore psychological empowerment, collaborative Crafting; psychological detachment, and related emotional disorders. The author’s effort is significant. Their findings are sound and the manuscript reads well and fluently.

Please allow me to suggest some additional related literature that I believe will further enrich the paper, namely:

Coelho, C. M., Suttiwan, P., Arato, N., & Zsido, A. N. (2020). On the nature of fear and anxiety triggered by COVID-19. Frontiers in psychology11, 3109.

Fernandez, A. B. (2020). Fear, anger and stress in medical decision-making during the COVID-19 outbreak. Covid Perspect Res. Rev1, 1-2.

Jørgensen, F. J., Bor, A., & Petersen, M. B. (2020). Compliance without fear: Predictors of protective behavior during the first wave of the COVID-19 pandemic. PsyArXiv. May19.

Kinman, G., Teoh, K., & Harriss, A. (2020). Supporting the well-being of healthcare workers during and after COVID-19. Occupational Medicine (Oxford, England), Vol. 70, pp. 294–296. https://doi.org/10.1093/occmed/kqaa096

Olesen, B., Gyrup, H. B., Troelstrup, M. W., Marloth, T., & Mølmer, M. (2020). Infection prevention partners up with psychology in a Danish Hospital successfully addressing staffs fear during the COVID-19 pandemic. Journal of Hospital Infection105(2), 377-378.

Reviewer 2 Report

Dear Authors, 

This study clarified the importance of psychological empowerment for preventing emotional disorder in health workers using appropriate analytical methods.

Readability is also fine. 

I would like to leave some comments.

Authors concluded that fear determined health workers inability to distance themselves from work. However, I feel that whether each health worker can distance appropriately from work largely depend on environment of working place more than individuals' fear of COVID-19. Can Authors explain the possible effect of how much work place is in a state of emergency, manpower on distancing work?

MInor

The term "COVID-19" sometimes appear as "Covid-19(lower case)." It is better to unify the case of the term through the manuscript.

A p-value could not be exactly <.000. Author should change this description (e.g. <.001).

Author should state generalizability in the limitation part due to sampling in sole institution, and low response rate. 

The situation of quarantine in Spain or the province in which this study was conducted as of July 2020 could help readers understand background of the situation at that time.

Reviewer 3 Report

This study tackles a relevant and timely issue, aiming at validating a measure of Covid-19-related worries in Spanish healthcare professionals. The methodology adopted is generally sound and appropriate. However, I have some concerns that I think should be addressed before the study is accepted for publication.

 Abstract

  • Lines 18-19 read “In various countries, the Fear of Covid-19 Scale has been shown to be a fast and reliable tool that helps prevent feelings of anxiety and depression”. Reformulate this sentence, as it seems to suggest that authors are referring to a tool for psychological interventions instead of a measurement tool, that can not be in itself used to prevent anxiety or depression.
  • Line 22 refers to the aim of adapting the Fear of Covid-19 scale but it’s not clear from the abstract what kind of adaptation is proposed.
  • I think citations should not be included in the abstract. 

Introduction

  • The acronym “PE” is only explained in the abstract. The full formulation “psychological empowerment” should be included in the main text as well at the first occurrence of “PE”.
  • Hypotheses should be justified more clearly. I also find slightly confusing the fact that each hypothesis is presented and justified separately. I would suggest to rearrange this section presenting first the literature from which the formulated hypotheses descend, then listing all of the hypotheses together.

Materials and Methods

  • Considering the adoption of SEM analysis, I would suggest to replace Cronbach’s alphas and correlations (Pearson?)with a CFA.
  • The paragraph 2.3 explains why consistent PLS was used instead of traditional PLS, but it is not totally clear why this approach was preferred over CB-SEM adopting e.g. the WLSMV estimator, which is specifically designed for ordinal variables as the ones in this study (I suppose it was because of the small sample size).
  • There are some typos (e.g. “valuation” instead of “evaluation”, “colinearity” instead of “collinearity”, “self-informed” instead of “self-report”)
  • Regarding the hypothesized model:
  • Why were direct effects of collaborative crafting on emotional disorders, fears of covid-19 and psychological detachment not included in the model? This choice does not allow to assess whether these are total or partial mediations, or to calculate total effects and how much of the total effect is explained by the mediation.
  • Why were emotional disorders considered as dependent on fear of Covid-19 and not on the same levels (both as mediators)? The causal direction does not seem clear enough, and this is troubling especially because data is cross-sectional. It seems reasonable that emotional disorders might affect fear of Covid-19 as well as the opposite.

Results

  • Lines 245-246 read: “Second, convergent validity was assessed, examining the loadings or simple correlations of indicators with their construct”. As convergent validity refers to the degree to which two constructs that should be theoretically related are in fact related, I don’t understand in which sense and how Authors have estimated it. What they refer to here seems more related with reliability than convergent validity. The strong association with emotional disorders seems like a better indicator of good convergent validity to me.
  • Line 287 refers to “postulated working hypotheses”. This is a contradiction in terms, as hypotheses are tested, while postulates are assumed to be true without proof.
  • Betas should have leading zeroes, as they can exceed 1. I think JCM also includes leading zeroes for p values.
  • “p < .000” means that the probability is negative. Did Authors mean “p < .001”?
  • Why were betas reported for some effects while t were reported for others?
  • I suggest to report coefficients with 3 decimal places
  • Lines 300-302 read: “The indirect effect of collaborative crafting on emotional disorders was also significant, through the mediation of PE (β = -.06, p < .05)”. Firstly, the exact p value should be reported. Secondly, is seems strange that such a small indirect effect is in fact statistically significant. Do authors think it is also practically significant enough to be discussed?

Discussion

  • The conclusion about the effects of fear of Covid-19 on emotional disorders is not warranted, as no theoretical reason was presented to justify the directionality of this association, especially considering the cross-sectional nature of the presented data

Round 2

Reviewer 3 Report

I commend the authors for quickly and thoroughly revising the manuscript. I only have two minor remarks to add, namely:

  • some p values are still not reported in their exact values as previously suggested (and as prescribed by the APA manual as well as most other guidelines for results reporting), e.g. lines 297 and 306. Only values smaller than .001 should be reported this way.
  • I find the way the heading of the 3.4.1-3.4.3 paragraphs confusing (e.g. "3.4.1.1. st stage: Configuration invariance"). Changing "1. st stage" to "First stage" might be a solution.

I wish the authors all the best with their future research.

Author Response

Dear reviewer,

We want to thank you for your valuable comments that have significantly strengthened and improved the manuscript.

We are very grateful.

We have underlined the modifications. We have changed p values ​​on lines 297 and 306.

We have also rectified subtitles by First, Second and Third.

Best regards

Marta Llorente-Alonso